# Reduction in the Allelopathic Potential of *Conocarpus erectus* L. through Vermicomposting

Sami ur Rehman [1], Zubair Aslam [1,*], Bandar S. Aljuaid [2], Rana Nadeem Abbas [1], Saqib Bashir [3], Munawar Hussain Almas [4], Tahir Hussain Awan [5], Korkmaz Belliturk [6], Wafa'a A. Al-Taisan [7], Samy F. Mahmoud [2] and Safdar Bashir [3,*]

1 Department of Agronomy, University of Agriculture, Faisalabad 38000, Pakistan
2 Department of Biotechnology, College of Science, Taif University, Taif 21944, Saudi Arabia
3 Department of Soil and Environmental Sciences, Ghazi University, Dera Ghazi Khan 32200, Pakistan
4 Directorate of Floriculture (Training & Research) Punjab, Lahore 54800, Pakistan
5 Department of Agronomy, Rice Research Institute, Kala Shah Kaku 39020, Pakistan
6 Department of Soil Science and Plant Nutrition, Faculty of Agriculture, Tekirdag Namık Kemal University, Tekirdag 59030, Turkey
7 Department of Biology, College of Science, Imam Abdulrahman Bin Faisal University, Dammam 31441, Saudi Arabia
* Correspondence: zauaf@hotmail.com (Z.A.); sabashir@gudgk.edu.pk (S.B.)

**Abstract:** The utilization of vermi-technology to reduce allelopathic effects is a cost-effective, efficient, and environmentally appropriate alternative to traditional chemical and mechanical methods. The current investigation was an effort to obtain vermicompost from *C. erectus* and its binary combination with soil and farmyard manure (FYM) using *E. foetida*. The pH, EC, organic carbon, macro and micro-nutrients, $CO_2$ emission, the average growth rate of the worms, number of worms, number of cocoons, and weight gained by earthworms were analyzed by standard methods. The present study also investigated the effect of produced vermicompost on the growth and yield of mung beans (*Vigna radiata* L). The maximum reduction in soil pH was observed (6.47) in vermicompost of *C. erectus* leaves, among other treatments. The highest N (1.86%), P (0.15%), and K (0.41%) contents were found in the vermicompost of *C. erectus* leaves + FYM. Similarly, the maximum plant height (36.00 cm) was achieved in vermicompost of *C. erectus* leaves + FYM compared to other treatments. The highest SPAD value was observed (56.37) when the soil was treated with vermicompost (*C. erectus* leaves + FYM) @ 5 t ha-1, followed by the treatment where vermicompost (*C. erectus* leaves + soil) @ 8 t ha-1 was applied. The soil amendment of vermicompost (*C. erectus* leaves + FYM) @ 5 t ha$^{-1}$ showed competitive results (in terms of the growth and yield parameters of mung beans) compared to other types of vermicompost and *C. erectus* biomass. This study has the potential to reduce the phytotoxicity of *C. erectus* biomass and transform it into a potent organic fertilizer through vermicomposting.

**Keywords:** *Conocarpus erectus*; vermicompost; allelopathy; organic fertilizer; mung bean

## 1. Introduction

*Conocarpus erectus* L. is grown in parks, roadsides, and field crops for esthetic value and the agro-forestry system but causes serious consequences on the environment and human health. One of the main effects on field crops is allelopathy—a chemical interaction in which one plant affects the development, growth, reproduction, and survival of adjacent plants by producing chemical compounds [1]. These chemical compounds (allelochemicals) are secondary metabolites of the acetate metabolic pathway and consist of phenolics; long-chain fatty acids; alkaloids; and derivatives of flavonoids, organic acids, and quinines. They may impede germination, respiration, photosynthesis, water relations, growth, and ion uptake [2]. Moreover, known sites of allelochemicals actions are cell division, photosynthesis, specific enzyme function and amino acid metabolism, and iron concentration.

Thus, allelopathy affects plant invasions and individual performance and reduces plant performance by 25% [3].

One of the challenges in this field of investigation is reducing allelochemicals in *C. erectus*. There is no effective method for reducing its allelopathic potential. The burning of *C. erectus* residues pollutes the environment and causes smog formation. Different methods of reducing the allelopathic effect are used, but vermicomposting seems to be the most feasible technology [4]. Vermicomposting is a biotechnological method of renovating biological waste into a nutrient-rich product called vermicast using earthworms. Vermicompost is peat-like material and has good structure, aeration, porosity, and high-water holding capacity [5].

Pulses are a chief source of protein and micronutrients such as iron [6], vitamin B, and several other minerals. Mung beans contain 20.97–31.32% protein [7] which is double the protein content of maize seeds (7–10%) [8] and is more than root crops [9]. Mung bean can be grown as a catch crop in the rice-wheat cropping system from April to June due to its short duration and restorative crops [10]. As it is a legume crop, it fixes atmospheric nitrogen, improves the fertility of the soil, and needs less irrigation [11]. In Pakistan, mung bean is grown on 186,700 hectares of land with 132,700 tons of production [12]. However, the yield potential of mung bean in Pakistan is low due to an imbalance in the plant's nutritional plan and loss of organic matter in the soil. The organic matter in soil has depleted due to the extensive use of synthetic fertilizers that adversely affect the soil's physical and chemical properties. Pakistani soils are deficient in organic matter, containing less than 1%. For the conservation of soil productivity, at least 2% organic matter is essential. Therefore, it is imperative to add soil organic amendments to restore organic matter loss in soil [13]. Among different organic amendments, vermicompost is the best organic fertilizer as it is free from any pathogens and is a source of nutrients such as nitrates, phosphates, exchangeable calcium, and soluble potassium [14]. Keeping in view the above scenarios, there is currently limited information regarding the reduction in the allelopathic potential of *Conocarpus erectus* L. through vermicomposting. The objectives of our study were (1) the reduction in the allelopathic *C. erectus* L. into nutrient-rich organic fertilizer and (2) the assessment of the influence of different rates and sources of vermicompost on mung bean growth and yield. This study can help to reduce the dependency on chemical fertilizers and take a step towards sustainable agriculture.

## 2. Materials and Methods

### 2.1. Experiment with C. erectus Vermicomposting Process

The treatments in this experiment were completed in vermicomposting pots of 35 cm × 15 cm at plant & microbial ecology lab, Department of Agronomy, University of Agriculture, Faisalabad (UAF). The leaves of *C. erectus* and farmyard manure were collected from the student research farm UAF and then shade-dried for 7 days. Then, these leaves were crushed into small pieces in a crusher at Biogas Plant, UAF. The crushed materials were kept for pre-composting in three pots, one pot for pure crushed leaves, and the remaining pots were filled with a mixture of crushed leaves with FYM and soil with a 50:50 ratio, respectively, for 20 days. $CO_2$ production in pre-composted material was calculated before the transfer of earthworms. After pre-composting of material, 40 earthworms (*Esenia fetida*) were placed into each of the three pots for 45 days, and moisture content was maintained at 65–70%. After 45 days, sieving was performed to separate vermicompost and earthworms from non-casted material. Newly prepared vermicompost was packed into zipper bags after shade-drying.

### 2.2. Physico-Chemical Analysis

The shade-dried vermicompost samples were physico-chemically analyzed. Total phenolic contents in vermicompost's extract were calculated using the Folin–Ciocalteu method [15]. A total of 100 g of pre-composting material was placed in a gas jar and was tightly locked and kept for 24 h; after, $CO_2$ gas was calculated using a Servomex 570A

oxygen analyzer and Servomex PA 404 gas analyzer (Servomex Company, Crowborough, Sussex, England) according to method described previously [16]. pH was calculated using a pH meter, and electrical conductivity (EC) was calculated with the help of a conductivity meter [17]. Nitrogen contents were measured using a Micro-Kjeldahl apparatus; phosphorus contents were measured using a spectrophotometer; and moisture, protein, and ash contents were measured using near-infrared spectroscopy (NIRS). Iron (Fe) and Zinc (Zn) in vermicompost samples were calculated using an Atomic Absorption Spectrophotometer (Hitachi Polarized Zeeman AAS, Z-8200, Japan).

### 2.3. Experiment to Assess the Effect of Soil Amendments of Vermicompost on Mung Bean

A pot experiment was conducted on mung bean (*Vigna radiata* L.) in open-field conditions at Student Research Farm, University of Agriculture, Faisalabad (UAF) during the summer of 2019. Soil used for the experiment was loam and collected at a 0–30 cm depth from the farm site. Soil has pH value of 8.3, electrical conductivity of 1.42 dSm$^{-1}$, nitrogen content of 0.045%, phosphorus content of 5.9 ppm, potassium content of 140 ppm, and organic matter content of 0.91%. The experiment was laid out in a completely randomized design (CRD) with seven treatments with three replicates, and the soil was clay loam. The applied treatments were T0 = Control; T1 = Vermicompost (*C. erectus* leaves) @ 5 t ha$^{-1}$; T2 = Vermicompost (*C. erectus* leaves) @ 8 t ha$^{-1}$; T3 = Vermicompost (*C. erectus* leaves + soil in 50:50 ratio) @ 5 t ha$^{-1}$; T4 = Vermicompost (*C. erectus* leaves + soil in 50:50 ratio) @ 8 t ha$^{-1}$; T5 = Vermicompost (*C. erectus* leaves + FYM in 50:50 ratio) @ 5 t ha$^{-1}$; and T6 = Vermicompost (*C. erectus* leaves + FYM in 50:50 ratio) @ 8 t ha$^{-1}$. Vermicompost was derived from different combinations of *C. erectus* leaves with soil and FYM. Then, this vermicompost was added to the pots containing 8 kg soil @ 5 t ha$^{-1}$ and 8 t ha$^{-1}$. PRI MUNG-2018 cultivar seed was purchased from Ayub Agriculture Research Institute (ARRI) Faisalabad, and sowing was completed. The manual weeding and optimum soil moisture of experimental pots were maintained throughout the experiment.

### 2.4. Biological Analysis

A total of 40 earthworms were added to different composting materials of *C. erectus* leaves. Number of earthworms and number of cocoons after vermicomposting of *C. erectus* leaves was calculated manually, whereas the weight of earthworms was measured by electric weight balance.

### 2.5. Plant Analysis to Observe the Effects of Soil Amendments of Vermicompost on Mung Bean Growth

Mung bean plants were harvested at maturity, and biochemical parameters of mung bean leaves for chlorophyll contents were measured at the flowering stage of the plant using a chlorophyll meter (SPAD-502). Agronomic characteristics of mung bean for emergence count per pot, plant height at harvest (cm), number of branches per plant, number of pods per plant, length of pods (cm), total grains per pod, 100-grain weight (g), nodules per plant, economic yield (kg/ha), biological yield (kg ha$^{-1}$), and root length (cm) at maturity were also measured.

### 2.6. Statistical Analysis

The experimental data were observed statistically using Fisher's analysis of variance technique. The least significant difference (LSD) test was used ($p < 0.05$) to compare significant treatment means. STATISTICA 8.0 was used to perform statistical analysis.

### 3. Results

The physico-chemical properties of different vermicomposts and control are presented in Table 1. The pH value of vermicompost of *C. erectus* leaves slightly decreased compared to the pH of *C. erectus* leaves. The EC values differ significantly ($p < 0.05$) in the case of *C. erectus* leaves and vermicompost of *C. erectus* leaves + FYM. Total organic carbon contents in all types of vermicomposts were reduced greatly compared to *C. erectus* leaves' biomass.

**Table 1.** Effect of vermicomposting on physico-chemical properties of *C. erectus* leaves with different combinations of FYM and soil. Values not marked with the same letter are significantly different at $p < 0.05$.

| Parameters | *C. erectus* Leaves | Vermicompost of *C. erectus* Leaves | Vermicompost of *C. erectus* Leaves + Soil | Vermicompost of *C. erectus* Leaves + FYM | LSD ($p < 0.05$) |
|---|---|---|---|---|---|
| pH | 6.85 A | 6.47 C | 6.56 BC | 6.69 B | 0.13 |
| EC | 4.32 C | 6.27 B | 6.95 AB | 7.54 A | 0.16 |
| Total organic carbon (%) | 39.23 A | 24.48 B | 21.36 C | 16.75 D | 1.92 |
| Nitrogen (%) | 1.35 D | 1.64 B | 1.56 C | 1.86 A | 0.06 |
| Phosphorus (%) | 0.01 C | 0.06 B | 0.07 B | 0.15 A | 0.01 |
| Potassium (%) | 0.26 C | 0.40 A | 0.29 B | 0.41 A | 0.01 |
| Zinc (ppm) | 21.50 D | 35.80 B | 30.00 C | 39.4 A | 1.12 |
| Iron (ppm) | 935.00 B | 1582.00 A | 1604.00 A | 1572.00 A | 36.98 |
| Copper (ppm) | 5.46 D | 11.60 C | 16.42 B | 19.46 A | 0.19 |
| Calcium (ppm) | 1918.20 D | 1981.50 C | 2223.50 B | 4085.80 A | 50.235 |

The nitrogen (N), phosphorus (P), and potassium (K) contents vary significantly ($p < 0.05$) in all treatments of vermicompost and plant biomass. However, the highest N, P, and K contents were found in the vermicompost of *C. erectus* leaves + FYM. This is because the nutrient contents in the final vermicompost depend on the initial nutrient status of the biomass. Zinc, copper, and calcium contents were significantly higher in vermicompost of *C. erectus* leaves + FYM compared to other vermicomposts and plant biomasses. Meanwhile, the iron content was found to be higher in the vermicompost of *C. erectus* leaves + soil. Vermicomposting of *C. erectus* leaves reduced the phenolic contents in all types of vermicompost compared to plant biomass. However, a higher reduction (31%) in phenolic contents was observed in the vermicompost of *C. erectus* leaves. $CO_2$ is one of the main gases that are released during the composting of organic wastes. The dynamics of $CO_2$ released during the pre-composting of *C. erectus* leaves are shown in Figure 1.

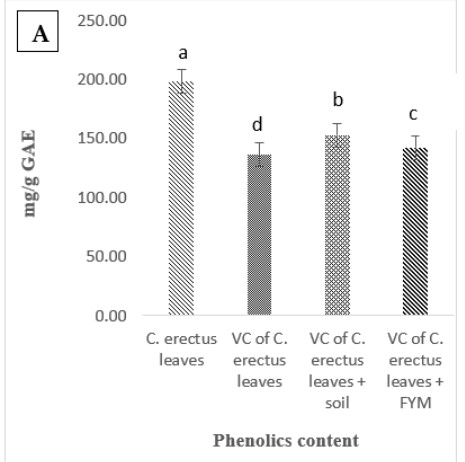
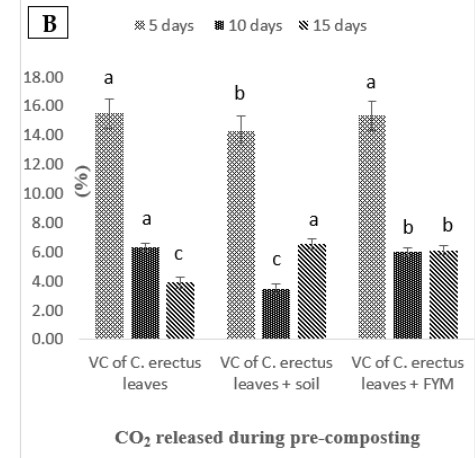

**Figure 1.** Effect of vermicomposting on phenolic contents (**A**) and $CO_2$ released (**B**) during the process. Different letters mean statistically significant differences, and bars represent the standard deviation among replicates of each treatment.

### 3.1. Effect of Vermicompost Treatments on Growth and Yield Parameters of Mung Bean

The effect of different treatments of vermicompost on the emergence count per pot, plant height, number of pods per plant, pod length, grains per pod, nodules per plant, root length, chlorophyll contents, 100-grain weight, biological yield, and economic yield is recorded in Figure 2. The results showed that all yield-promoted parameters were significantly influenced by the application of vermicompost. The highest emergence count (96.66%) was detected in T5, where vermicompost (*C. erectus* leaves + FYM) was applied at 5 t ha$^{-1}$, followed by T6 (90.00%), where it was applied at 8 t ha$^{-1}$, compared to the control treatment (78.33%), where no vermicompost was applied. Maximum plant height (36.00 cm) was achieved in T5 compared to T6 and T3 treatments.

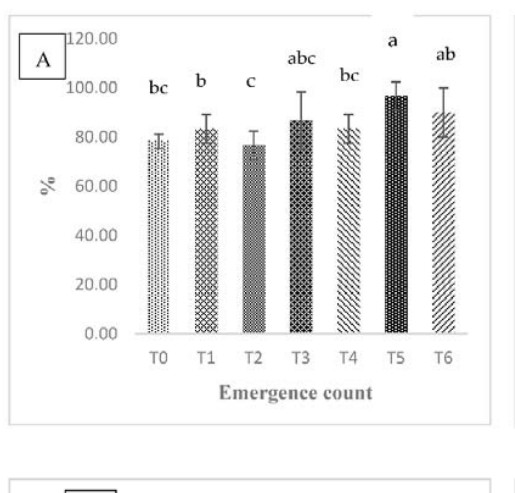
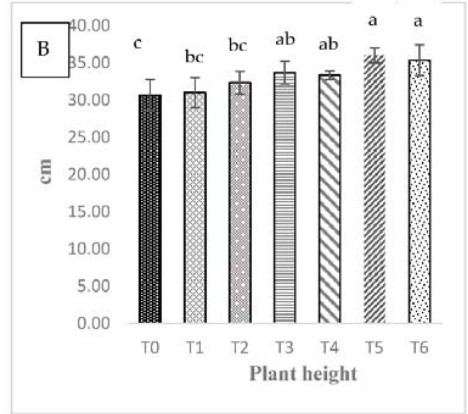
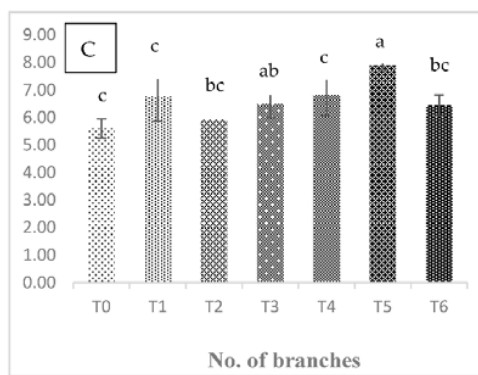
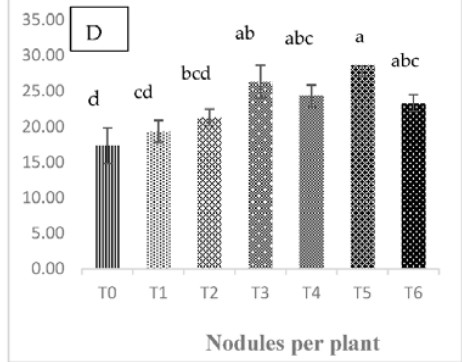
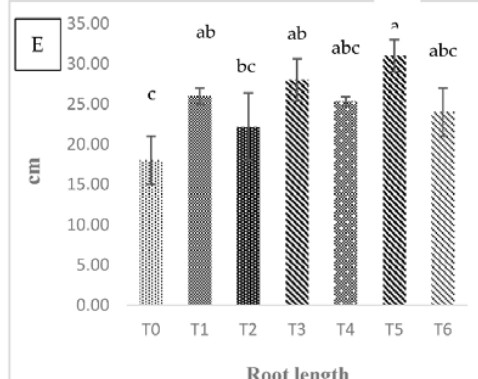
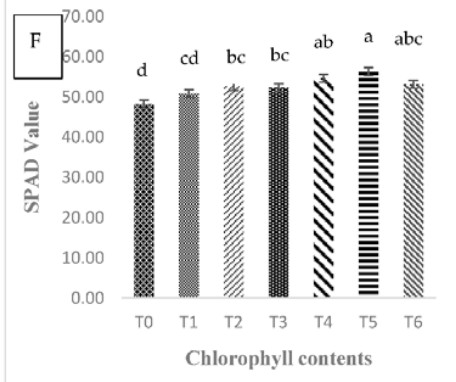

**Figure 2.** Changes in the growth characteristics of mung bean after the soil application of vermicompost. Different letters mean statistically significant differences and bars represent the standard deviation among replicates of each treatment. (**A**) Emergence count, (**B**) Plant Height, (**C**) No. of branches, (**D**) Nodules per plant, (**E**) Root length, (**F**) Chlorophyll content.

The average number of branches per plant was increased up to 30% and 18% with the application of vermicompost (*C. erectus* leaves + FYM) @ 5 t ha$^{-1}$ and vermicompost (*C. erectus* leaves + soil) @ 8 t ha$^{-1}$, respectively, compared to control treatment. The nodules per plant and root length of mung bean were observed to be 40% and 42% more in vermicompost (*C. erectus* leaves + FYM) @ 5 t ha$^{-1}$ compared to the control. The fastest method to estimate the chlorophyll content is the use of a chlorophyll meter (SPAD-502). The SPAD value of all the treatments ranged between 48.21 and 56.37, and the highest value was obtained in the treatment with vermicompost (*C. erectus* leaves + FYM) @ 5 t ha$^{-1}$, followed by the treatment where vermicompost (*C. erectus* leaves + soil) @ 8 t ha$^{-1}$ was applied (Figure 2).

The yield-contributing parameters of mung bean differ significantly among all vermicompost treatments, as shown in Figure 3. The number of pods per plant increased up to 40% with soil amendment of vermicompost (*C. erectus* leaves + FYM) @ 5 t ha-1 in comparison with the control treatment. The application of vermicompost (*C. erectus* leaves + FYM) @ 5 t ha$^{-1}$ enhanced the pod length (29%), grains per pod (28%), and 100-grain weight (22%) of mung bean in contrast to no application of vermicompost. The economical and biological yield of mung beans was boosted to 59% and 34% with the application of vermicompost (*C. erectus* leaves + FYM) @ 5 t ha$^{-1}$, respectively.

The average number of branches per plant increased to 30% and 18% with the application of vermicompost (*C. erectus* leaves + FYM) @ 5 t ha$^{-1}$ and vermicompost (*C. erectus* leaves + soil) @ 8 t ha$^{-1}$, respectively, compared to the control treatment. The nodules per plant and root length of mung bean were observed to be 40% and 42% more in vermicompost (*C. erectus* leaves + FYM) @ 5 t ha$^{-1}$ compared to the control. The fastest method to estimate the chlorophyll content is the use of a chlorophyll meter (SPAD-502). The SPAD value of all the treatments ranged between 48.21 and 56.37, and the highest value was obtained in the treatment with vermicompost (*C. erectus* leaves + FYM) @ 5 t ha$^{-1}$, followed by the treatment where vermicompost (*C. erectus* leaves + soil) @ 8 t ha-1 was applied (Figure 2).

The yield-contributing parameters of mung bean differ significantly among all vermicompost treatments, as shown in Figure 3. The number of pods per plant increased to 40% with soil amendment of vermicompost (*C. erectus* leaves + FYM) @ 5 t ha-1 in comparison with the control treatment. The application of vermicompost (*C. erectus* leaves + FYM) @ 5 t ha$^{-1}$ enhanced the pod length (29%), grains per pod (28%), and 100-grain weight (22%) of mung beans in contrast to no application of vermicompost. The economical and biological yield of mung beans were boosted up to 59% and 34% by the application of vermicompost (*C. erectus* leaves + FYM) @ 5 t ha$^{-1}$, respectively.

### 3.2. Effect of Vermicomposting of C. erectus Leaves on Earthworms

No earthworm mortality was observed during the composting process of *C. erectus* leaves with a different combination of FYM and soil. However, the number of earthworms increased with vermicomposting. However, during the harvesting of vermicompost, the number of cocoons increased. The highest number of earthworms, number of cocoons, average weight of earthworms, and growth rate of the worms significantly increased during the vermicomposting of *C. erectus* leaves + FYM, as shown in Table 2.

**Table 2.** Effect of vermicomposting of *C. erectus* leaves with different combinations of FYM and soil on earthworms population. Values not marked with the same letter are significantly different at $p < 0.05$.

| Substrate | Number of Earthworms | Number of Cocoons | Average Weight of Earthworm (mg) | Growth Rate/Worm/Day (mg) |
|---|---|---|---|---|
| *C. erectus* leaves | 43.00 B | 12.00 B | 686.67 B | 7.11 B |
| *C. erectus* leaves + soil | 47.33 AB | 16.00 AB | 836.67 AB | 9.33 A |
| *C. erectus* leaves + FYM | 55.00 A | 23.00 A | 940.00 A | 11.11 A |
| LSD ($p < 0.05$) | 10.33 | 9.98 | 198.34 | 1.99 |

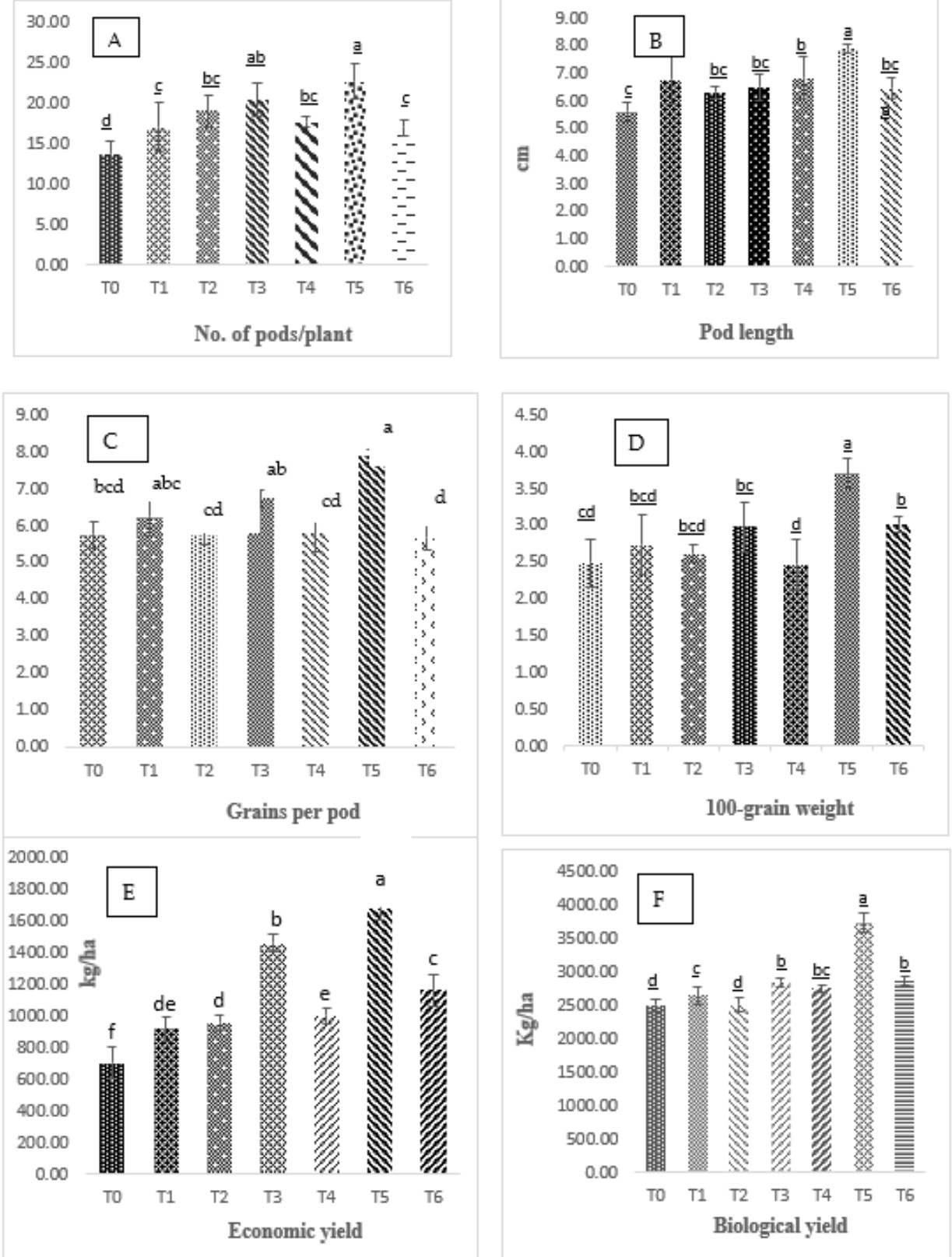

**Figure 3.** Changes in the yield-contributing attributes of mung bean after the soil application of vermicompost. Different letters mean statistically significant differences and bars represent the standard deviation among replicates of each treatment. (**A**) No. of Pod per plant, (**B**) Pod length, (**C**) Grain per pod, (**D**) 1000 grain weight, (**E**) Economic Yield, (**F**) Biological yield.

## 4. Discussion

Allelopathy in agriculture has a detrimental effect on the target plant, soil, microbe, and environment. Allelochemicals can trigger soil sickness and auto-toxicity and have a negative impact on crop growth [18]. A plant with phenolic contents has phytotoxicity toward specific plant species and the environment [19]. The results of the present studies unveiled that *C. erectus* residues are rich in phenolic compounds, which indicates their allelopathic potential against crop plants. The vermicomposting of different combinations of *C. erectus* leaves with FYM and soil significantly reduced the phenolic contents of *C. erectus* leaves. The highest reduction in phenolic contents was observed in vermicompost of *C. erectus* leaves with FYM, as FYM acts as a buffer to allelochemicals [20]. Our results are supported by [21,22], who conducted several experiments on the reduction in allelopathy of *Parthenium hysterophorus* L. by vermicomposting. There was also an improvement in the average weight of worms, number of earthworms, number of cocoons, and growth rate of earthworms in different combinations of *C. erectus* leaves with FYM and soil. However, the highest earthworm growth parameters were observed in *C. erectus* leaves and FYM composting material. Different factors are important to earthworm growth and reproduction, but the type of composting material plays a significant role in earthworm growth and reproduction [23]. No earthworm mortality was observed during the vermicomposting of *C. erectus* leaves. The death of earthworms is the result of an anaerobic condition developed after two weeks of the vermicomposting process of waste material [24]. In our experiment, all the vermicomposting material was pre-composted before the addition of earthworms into the material. Therefore, all the toxic gases produced were released from the material during the pre-composting of *C. erectus* leaves. Consequently, pre-composting is considered the best practice to avoid earthworm mortality. The growth rate is used as a comparative index of worms to compare their growth in different wastes. The highest growth rate was observed in *C. erectus* leaves + FYM material compared to *C. erectus* leaves and *C. erectus* leaves + soil material (Table 2).

$CO_2$ is one of the main gases produced during the composting of *C. erectus* leaves and its mixture with soil and FYM. The release of $CO_2$ depends on the temperature of composting material. More $CO_2$ emission was observed in the initial stage of composting, characterized by higher temperatures and intensive biodegradation of waste material.

During vermicomposting, there was a significant increase in biochemical characteristics such as N, P, K, Zn, Fe, and other micronutrients in the final vermicompost compared to the starting feed material. The increase in nitrogen contents of vermicompost is due to the mineralization of *C. erectus* biomass and earthworms stabilizing nitrogen, excreta, enzyme, mucus, and other enzymes [25,26]. The increased phosphorus contents in the final vermicompost is due to the role of phosphorus-solubilizing bacteria in the earthworm gut [27]. Total potassium increased in all types of vermicompost compared to *C. erectus* biomass. However, total potassium increased by 37% in *C. erectus* leaves + FYM vermicompost in comparison with uncomposted material. Our findings are supported by [28], who reported that the total potassium increased 7–10% percent during vermicomposting. This increase in potassium contents is due to speedy mineralization by microbial action [29]. Zinc, iron, copper, and calcium content was lowered in fresh *C. erectus* leaves compared to vermicomposted *C. erectus* leaves with soil and FYM combinations, and these findings agreed with the studies of [30]. The decrease in pH of vermicompost might be due to microbial activity and the elimination of volatile ammonia during vermicomposting. The EC of vermicompost increased compared to *C. erectus* leaves. The increase in EC might be due to organic matter loss and the release of mineral salts in available forms, such as ammonium, phosphate, and potassium [31].

In the present study, the highest emergence count (96.66%) was detected in vermicompost-treated plants, as vermicompost is rich in organic chemicals and is possible to contain compounds that boost germination [32]. Maximum plant height (36.00 cm) was gained in T5, where vermicompost (*C. erectus* leaves + FYM) was applied at 5 t ha$^{-1}$. These findings were supported by [33], who described that plant height boosted with

vermicompost amendment in soil. Vermicompost holds a combination of macro and micro-nutrients, and the uptake of these nutrients has a positive effect on nutrition, plant growth, chlorophyll contents, and the photosynthesis of leaves [34]. The authors of [35] detected that the biomass and plant height of sorghum were significantly higher with the application of earthworm casts and soil mixture. The maximum number of pods (22.66) were recorded with vermicompost (*C. erectus* leaves + FYM) @ 5 t ha$^{-1}$. The authors of [36] described that the use of humic substance and phosphorus increased the number of flowers and pods per plant compared to the control. Our findings are in harmony with [37], who noted that the highest pod length of bush bean, long, and winged bean—10.75 cm, 30 cm, and 22.27 cm, respectively—were found in treatment with vermicompost applied compared to the control. The data on the number of grains per pod of mung bean as influenced by vermicompost prepared using different ratios of conocarpus leaves, soil, and FYM and applied at 5 t ha-1 and 8 t ha$^{-1}$ are shown in Figure 3. The authors of [38] recorded the highest number of grains per pod from a combination of MoP @ 20 kg ha$^{-1}$ and vermicompost @ 8 t ha$^{-1}$.

Leguminous crops, such as mung beans, have root nodules that serve as a home for rhizobia (nitrogen-fixing bacteria). The maximum number of nodules (29.00) was shown where vermicompost (*C. erectus* leaves + FYM) was applied @ 5 t ha$^{-1}$, and these findings are supported by [39,40]. The application of vermicompost (*C. erectus* leaves + FYM) proliferates the root length and chlorophyll contents of mung beans. This might be due to the beneficial effect of vermicompost in root proliferation with higher carbon content [41]. The improvement in chlorophyll content with the application of vermicompost was also detected by [42]. In the present experiment, vermicompost affected the 100-grain weight of mung bean, and statistically, the results were highly significant. These findings are in parallel to the results of [43] in mung beans and [44] in groundnuts. The authors of [45] reported that soil amendment of vermicompost @ 7.5 t ha$^{-1}$ increased the dry matter, plant height, dry weight of root nodules, chlorophyll contents, and biological yield of mung beans. Data related to the economic yield of mung beans at harvest are described in Figure 3. These results are supported by Rajkhowa et al. (2000), who disclosed that the grain yield was highly significant in crops receiving vermicompost @ 5t ha$^{-1}$ + NPK at the recommended dose compared to treatment receiving NPK alone. The current experiment demonstrated that vermicomposting reduced the allelochemicals and converted *C. erectus* leaves into benign fertilizer. The positive outcomes of this study may be used to reduce the allelopathic effect of different crop residues and the production of organic fertilizer.

## 5. Conclusions

The present study shows that the allelochemicals in *C. erectus* leaves are reduced through the vermicomposting process. The highest reduction (31%) in allelochemicals is found with the vermicomposting of *C. erectus* leaves + FYM. However, the soil amendment of vermicompost (*C. erectus* leaves + FYM) @ 5 t ha$^{-1}$ produced competitive results (in terms of growth and yield parameters of mung bean) compared to other types of vermicompost. Thus, vermicomposting can be a fit method for reducing the allelopathic effect of *C. erectus* leaves, and non-toxic fertilizer is obtained.

## 6. Future Prospective

The results of the current study lay the foundation for future research examining the potential use of vermicomposting to reduce the allelopathic effect of different crops. More research into the large-scale use of these vermicomposts and their impacts on crop development and production is recommended.

**Author Contributions:** Conceptualization, S.u.R. and S.B. (Saqib Bashir); methodology, B.S.A. and Z.A.; software, R.N.A.; validation, S.B. (Safdar Bashir), M.H.A. and T.H.A.; formal analysis, Z.A.; investigation, S.B. (Saqib Bashir); resources, K.B.; data curation, Z.A.; writing—original draft preparation, S.u.R.; writing—review and editing, S.B. (Safdar Bashir); visualization, Z.A.; super-vision, M.H.A.; project administration, W.A.A.-T., S.F.M. and S.B. (Safdar Bashir); funding acqui-sition, Z.A. All authors have read and agreed to the published version of the manuscript.

**Funding:** The authors are grateful to HEC for funding this research through the following HEC projects: "Vermicomposting: A resourceful organic fertilizer to improve agriculture production and soil health" (NRPU-HEC project no. 7527/Punjab/NRPU/R&D/HEC/2017) and "Vermicomposting: An Agricultural Waste Management Technology" (project vide letter no. (Ph- II-MG-9)/PAKTURK/R&D/HEC/2018, though Pak-Turk Researchers Mobility Grant Program Phase- 2).

**Institutional Review Board Statement:** Not applicable.

**Informed Consent Statement:** Not applicable.

**Data Availability Statement:** Not applicable.

**Acknowledgments:** Taif University Researchers Supporting Project number (TURSP-2020/245), Taif University, P.O. Box 11099, Taif 21944, Saudi Arabia. The authors are grateful to HEC for funding this research through the following HEC projects: "Vermicomposting: A resourceful organic fertilizer to improve agriculture production and soil health" (NRPU-HEC project no. 7527/Punjab/NRPU/R&D/HEC/2017) and "Vermicomposting: An Agricultural Waste Management Technology" (project vide letter no. (Ph- II-MG-9)/PAKTURK/R&D/HEC/2018, though Pak-Turk Researchers Mobility Grant Program Phase- 2).

**Conflicts of Interest:** The authors declare no conflict of interest.

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
