# Peer review of "Reduction in the Allelopathic Potential of Conocarpus erectus L. through Vermicomposting"

_sustainability, doi:10.3390/su141912840_

Round 1

Reviewer 1 Report

I truly appreciate the authors for their thorough investigation into such a crucial and modern topic. The whole article, particularly the way in which concepts are presented while being supported by rigorous theoretical analysis and empirical research, significantly contributes to the body of literature already in existence. To improve the quality of the manuscript even more, it is suggested that the writers add (or change) the following things:

1. The amount of literature support (in the form of citations) looks to be substantially smaller throughout the manuscript. Please consider providing appropriate and current literary support wherever possible.

2. The levels of significance in Tables 1 and 2 should be shown using the same rules that are used in the literature. For example, use ** and * for each value to show its significance at different levels. Please also use "Table" instead of "Tab" in the text.

3. It would also be beneficial for the readers if the author could add limitations and strengths of the study.

4. Authors should try to shorten the table captions.

Author Response

I truly appreciate the authors for their thorough investigation into such a crucial and modern topic. The whole article, particularly the way in which concepts are presented while being supported by rigorous theoretical analysis and empirical research, significantly contributes to the body of literature already in existence. To improve the quality of the manuscript even more, it is suggested that the writers add (or change) the following things:

Response: Authors appreciate the reviewer’s impression for the manuscript.

  1. The amount of literature support (in the form of citations) looks to be substantially smaller throughout the manuscript. Please consider providing appropriate and current literary support wherever possible.

Response: Authors are thankful to reviewer for his valuable suggestion. Citations are updated in the whole document

  1. The levels of significance in Tables 1 and 2 should be shown using the same rules that are used in the literature. For example, use ** and * for each value to show its significance at different levels. Please also use "Table" instead of "Tab" in the text.

Response: Thanks to the reviewer for valuable comment. The suggestions are incorporated. Values not marked with the same letter are significantly different

  1. It would also be beneficial for the readers if the author could add limitations and strengths of the study.

 Response: The strengths of the study are are mentioned in lines 72-76 which is the development of organic fertilizers using allelopathic C. erectus L. The limitation may be the impact of vermicompost produced from other allelopathic plants might be variable due to variable physico chemical composition of the plants. So, more research can confirm this behavior.

  1. Authors should try to shorten the table captions.

Response: Thanks to reviewer for the suggestion the table caption are modified

Reviewer 2 Report

Reduction of allelopathic potential of Conocarpus erectus L. through vermicomposting

Review

Introduction

I propose to extend the information on the basis of the literature on the origin and occurrence of Conocarpus erectus L. and mung beans. It would also be good to give a taxonomy of species. Is it not an exaggeration to write that "C. erectus residues pollute the environment and causes smog formation" (line 45)? If the problem occurs on a large scale, I propose to refer to the relevant literature at this point.

Materials and Methods

When vermicompost was produced? (line 76-88)? The text only says that mung beans were grown in 2019 (line 104). The text says "CO2 production in pre-composted material was calculated before the transfer of earthworms" (line 83-84). Please explain how it was calculated and refer to the methodology described in the literature. Please provide information whether mature earthworms were used in the experiment and where they came from. Were they harvested or bought somewhere from their own breeding? What was their starting biomass (line 84-85; 120-123)?

There is no information on how many replicates the entire experimental setup was performed. How long was the mung bean experiment (line 118)? Line 122 - I propose to add a reference to the literature on the "hand sorting method".

There is no information on how the results were presented (line 134-136). Were they arithmetic means and standard deviations or standard errors? Please explain why Fisher's analysis of variance technique was used? Have you checked whether the data is normally distributed and whether the variances are homogeneous? I propose to write what program was used to perform the statistical analyzes.

Results

Are you sure "The pH value of vermicompost of C. erectus leaves decreased slightly than pH of C. erectus leaves" (line 139)? The difference is significant, which is confirmed by the statistical markings in the table. "Values ​​not marked with the same letter are significantly different at p <0.05" - it can be written more simply, eg "different letters mean statistically significant differences" (line 144-145). Maybe marking the elements with the same symbols in the table is enough?

Please add "plant biomass" (line 147).

Please re-analyze the statistical determinations. Is it sure "... the iron content was found higher in vermicompost of C. erectus leaves + soil" (line 152).

I propose to mark particular graphs, eg "a" and "b", eg "Effect of vermicomposting on phenolic content (a) and CO2 released during the precess (b)" (line 159-160). Similarly in Figure 2.

There are 6 graphs in Figure 2 and 11 are listed in the text. It is mixed up, other parameters are listed in the text and others are arranged in the Figures (line 163-166). Line 170 - Please check again the statistical markings in Figure 2. Line 172 - In the text it says "branches per plant", in the chart "no of branches" please check it. Line 172-180 the description in the text does not match the diagram. Please check it again.

There is no description of Figure 3. From the text it appears that it is Figure 3. Line 218 - discard the sentence: "At the initial stage of vermicomposting, no cocoons were added.

Line 219 -" a lot of cocoons "is not a good term because there were only a dozen of them. Maybe it would be better to put mature and young specimens separately in the table - it will be better to see the lack of mortality (line 227-229).

References

Most of the old literature is included in the text. Please enrich the article also with newer literature items.

Author Response

Introduction

I propose to extend the information on the basis of the literature on the origin and occurrence of Conocarpus erectus L. and mung beans. It would also be good to give a taxonomy of species. Is it not an exaggeration to write that "C. erectus residues pollute the environment and causes smog formation" (line 45)? If the problem occurs on a large scale, I propose to refer to the relevant literature at this point.

 Response: Authors are thankful to reviewer comment and appreciate their effort to review this manuscript thoroughly. The statement in line 45 is just an example and it is normally practiced burning urban tree waste in developing countries and it causes smog in winter. Following more suitable citation is added

Raza, Waseem, Saad Saeed, Hammad Saulat, Hajera Gul, Muhammad Sarfraz, Christian Sonne, Z-H. Sohn, Richard JC Brown, and Ki-Hyun Kim. "A review on the deteriorating situation of smog and its preventive measures in Pakistan." Journal of Cleaner Production 279 (2021): 123676.

Materials and Methods

When vermicompost was produced? (line 76-88)? The text only says that mung beans were grown in 2019 (line 104). The text says "CO2 production in pre-composted material was calculated before the transfer of earthworms" (line 83-84). Please explain how it was calculated and refer to the methodology described in the literature. Please provide information whether mature earthworms were used in the experiment and where they came from. Were they harvested or bought somewhere from their own breeding? What was their starting biomass (line 84-85; 120-123)?

Response: Line 81-91 describes method of vermicompost production. For CO2 Analysis, 100 g pre-composting material was placed in a gas jar and was tightly locked and kept for 24 hours, after that CO2 gas was calculated by a Servomex 570A oxygen analyzer and Servomex PA 404 gas analyzer (Servomex Company, Crowborough, Sussex, England) as used by (Vikman, Itävaara, & Poutanen, 1995).

Vikman, M., Itävaara, M., & Poutanen, K. (1995). Measurement of the biodegradation of starch-based materials by enzymatic methods and composting. Journal of environmental polymer degradation, 3(1), 23-29.

There is no information on how many replicates the entire experimental setup was performed. How long was the mung bean experiment (line 118)? Line 122 - I propose to add a reference to the literature on the "hand sorting method".

Response: Thanks to reviewer. The information about replicates is added in line 115. The experiment time for mung bean crop from sowing till maturity was 105 days. Manual weeding is normally practiced and commonly known so mentioning with references is not imporatant

There is no information on how the results were presented (line 134-136). Were they arithmetic means and standard deviations or standard errors? Please explain why Fisher's analysis of variance technique was used? Have you checked whether the data is normally distributed and whether the variances are homogeneous? I propose to write what program was used to perform the statistical analyzes.

 Response: The experimental data observed statistically by using Fisher’s analysis of variance tech-nique. The least significant difference (LSD) test was used (p<0.05) to compare significant treatment means. Standard error was calculated to check variability among replicates. Statistica.8.0 was used to perform all statistical analysis.

Results

Are you sure "The pH value of vermicompost of C. erectus leaves decreased slightly than pH of C. erectus leaves" (line 139)? The difference is significant, which is confirmed by the statistical markings in the table. "Values ​​not marked with the same letter are significantly different at p <0.05" - it can be written more simply, eg "different letters mean statistically significant differences" (line 144-145). Maybe marking the elements with the same symbols in the table is enough?

Response: Yes the pH observation was same as mentioned in the manuscript this might be due to effect of vermicomposting during digestion of leaves and production of more acidic compounds in earthworm guts. The statement is replaced with suggested statement.

Please add "plant biomass" (line 147).

Response: Thanks, the suggestion is incorporated

Please re-analyze the statistical determinations. Is it sure "... the iron content was found higher in vermicompost of C. erectus leaves + soil" (line 152).

Response: Thanks to reviewer for the concern the data was checked carefully also at the time of analysis we had the same observation and samples were re-analyzed and same findings were observed.

I propose to mark particular graphs, eg "a" and "b", eg "Effect of vermicomposting on phenolic content (a) and CO2 released during the precess (b)" (line 159-160). Similarly in Figure 2.

Response: Suggestion is incoorporated

There are 6 graphs in Figure 2 and 11 are listed in the text. It is mixed up, other parameters are listed in the text and others are arranged in the Figures (line 163-166). Line 170 - Please check again the statistical markings in Figure 2. Line 172 - In the text it says "branches per plant", in the chart "no of branches" please check it. Line 172-180 the description in the text does not match the diagram. Please check it again.

Response:There are three figures in the whole manuscript. The statement is added to make the figures clearer as “Different letters mean statistically significant differences and bar represent the standard deviation among replicates of each treatment”

There is no description of Figure 3. From the text it appears that it is Figure 3. Line 218 - discard the sentence: "At the initial stage of vermicomposting, no cocoons were added.

Response: Thanks to reviewer the suggestion is in corporated

Line 219 -" a lot of cocoons "is not a good term because there were only a dozen of them. Maybe it would be better to put mature and young specimens separately in the table - it will be better to see the lack of mortality (line 227-229).

 Response: Thanks to reviewer the term is modified line 244. We have mentioned the growth rate of earthworm and only this date was recorded and thanks to reviewer in future this observation can be documented

References

Most of the old literature is included in the text. Please enrich the article also with newer literature items.

Response: The reference list is updated
